# Formation of Periodic Nanoridge Patterns by Ultrashort Single Pulse UV Laser Irradiation of Gold

**DOI:** 10.3390/nano10101998

**Published:** 2020-10-10

**Authors:** Andreas Blumenstein, Martin E. Garcia, Baerbel Rethfeld, Peter Simon, Jürgen Ihlemann, Dmitry S. Ivanov

**Affiliations:** 1Laser-Laboratorium Göttingen e.V., Hans-Adolf-Krebs-Weg 1, 37077 Göttingen, Germany; peter.simon@llg-ev.de; 2Physics Department, University of Kassel, Heinrich-Plett-Str. 40, 34132 Kassel, Germany; garcia@physik.uni-kassel.de; 3Department of Physics and OPTIMAS Research Center, Technical University of Kaiserslautern, Erwin-Schrödinger-Str. 46, 67663 Kaiserslautern, Germany; rethfeld@physik.uni-kl.de; 4Quantum Electronics Division, Lebedev Physical Institute, 119991 Moscow, Russia

**Keywords:** periodic nanostructures, molecular dynamics, two-temperature model, laser material processing, ultrashort laser pulses, laser ablation, laser interference ablation

## Abstract

A direct comparison of simulation and experimental results of UV laser-induced surface nanostructuring of gold is presented. Theoretical simulations and experiments are performed on an identical spatial scale. The experimental results have been obtained by using a laser wavelength of 248 nm and a pulse length of 1.6 ps. A mask projection setup is applied to generate a spatially periodic intensity profile on a gold surface with a sinusoidal shape and periods of 270 nm, 350 nm, and 500 nm. The formation of structures at the surface upon single pulse irradiation is analyzed by scanning and transmission electron microscopy (SEM and TEM). For the simulations, a hybrid atomistic-continuum model capable of capturing the essential mechanisms responsible for the nanostructuring process is used to model the interaction of the laser pulse with the gold target and the subsequent time evolution of the system. The formation of narrow ridges composed of two colliding side walls is found in the simulation as well as in the experiment and the structures generated as a result of the material processing are categorized depending on the range of applied fluencies and periodicities.

## 1. Introduction

Since the beginning of ablative laser materials processing, the fabrication of periodic surface patterns has always been a subject of interest. Besides the spontaneous development of periodic ripple structures (LIPSS) [1,2,3] and surface plasmon-polariton (SPP)-induced structures under the controlled conditions [4,5,6], the beam interference concepts have been applied to create deterministic patterns with a predefined period and control of the surface profile [7,8]. For moderate period sizes in the range of a few µm to tens of µm, mostly nanosecond pulsed lasers have been applied [9,10,11,12]. If shorter periods in the µm or sub-µm range are desired, especially in case of metals and semiconductors, ultrashort pulses of picosecond or femtosecond duration are used [13,14,15]. The latter have the advantage that thermal diffusion of the absorbed energy and thus the extent of the heat affected zone can be minimized enabling precise patterning [16,17]. However, in many cases it has been observed that the local ablation depth does not correlate with the local laser fluence in a simple way as it could be expected from large-area ablation experiments, where a monotonously increasing ablation depth with increasing fluence is observed. Instead, the formation of bumps, voids, ridges, and droplets is observed, indicating that on this scale, other contributions like lateral material movement have to be considered [18]. Irradiation with a line pattern leads to grooves with protruding side walls [19]. Thin films disintegrate and form nanowires and nanodots [20,21] or nanospikes [22]. Thermocapillary and Marangoni effects as well as Rayleigh–Plateau instability have been identified to influence the structure formation and its disintegration process [23,24]. Later on, however, the relaxation of the laser-induced stresses, generated in the vicinity of the target’s surface as a result of the laser heating, has been determined as a main driving mechanism responsible for the nanostructures growth [25,26,27]. The specifics of the final nanostructures shape depend on the material as well as on the laser parameters, the dynamical changes of the reflectivity of the material [28,29], and the geometric parameters of the irradiation pattern.

In this study, we investigate the formation of structure on gold irradiated by an UV ultrashort pulse at a 248-nm wavelength with a periodic line pattern with periods of 270–500 nm. Experimentally obtained results are directly compared to simulations based on a hybrid atomistic-continuum model, where the atomic motion is described within a molecular dynamics (MD) approach, the effect of free carriers is considered with a diffusion differential equation describing the temperature dynamics of electrons, and the energy exchange between electrons and the lattice is treated as in the Two Temperature Model (TTM) [30]. With such combined MD-TTM approach [31], the mechanisms of ultrashort laser pulse melting, spallation, and ablation of mono- and poly-crystalline metal targets [32], nanostructuring in air [33] and under water confinement [34,35], as well as laser-induced generation of nanoparticles in different media [36] have been successfully studied. The results of simulations are analyzed and the target’s evolution is characterized via dynamic evolution of thermophysical dynamics quantities such as temperature, pressure, density, and the lift-off velocity. The strength and accuracy of the derived mechanisms are verified with the experimental measurements and observations. Such a methodology allows for extracting the mechanisms of the nanostructures formation and manipulating with them targeting the problem of generation of periodic patterns with predesigned morphology. The obtained structures are classified into four different fluence regimes resulting in (1) surface swelling, (2) internal void formation including structures growth, (3) surface walls formation, and (4) broad melting. The first two lower fluence regimes around the material’s modification threshold, corresponding to the onset of surface melting, sub surface voids generation and subsequent formation of periodic lines were described in [37]. The higher fluence regimes, around and above the ablation threshold of 210 mJ/cm^2^ [38], resulting in the formation of steep ridges and broad melting, including the process of generation of a subsurface layer (here 100 nm depth), with polycrystalline structures are studied in the present paper. The nanostructuring processes are well described by the simulations that match the experimental results for different irradiation conditions with great detail.

## 2. Experiment

The surface structuring experiments are performed by single pulse exposure using a combination of interference and mask projection to form a large area of sinusoidal intensity distribution on the sample surface (Figure 1). The used laser system consists of a frequency tripled Ti:Sa system (Coherent Libra) seeding a KrF excimer amplifier (LLG TwinAmp) producing pulse energies of up to 40 mJ at a wavelength of 248 nm and a repetition rate of 10 Hz. The pulse length of 1.6 ps is measured by using a frequency resolved optical gating (FROG) trace obtained with a self build device described in Ref. [39]. Single pulses are selected by a shutter. The used pulse energy is set by a variable attenuator. The flat top central part of the laser beam illuminates the mask comprising a square aperture of 1 mm edge length in contact with a Cr-on-quartz grating of lines and spaces with 25 µm period and duty cycle 0.5. The demagnification of the Schwarzschild objective used for mask imaging is adjusted to be 25x. Consequently, the edge length of the overall projected area on the workpiece is 40 µm. Since all diffracted orders emerging from the transmission grating except the two first orders are blocked, the image is formed via two-beam interference, resulting in a sinusoidal intensity distribution across the 40 µm area. The spatial frequency of this modulation is doubled by blocking the zeroth order diffraction, thus resulting in a period of 500 nm on the workpiece (instead of a pitch size of 1 µm based on the nominal demagnification). In order to obtain other periods, the mask period and objective demagnification have been changed accordingly. For the 350-nm structure a 36x objective and a 25-µm grating, and for the 270-nm period a 74x objective and a 40-µm grating were used. A schematic view of the intensity distribution is shown in Figure 1b. The purpose of the lens with f = 1000 mm in the setup is to reduce the size of the beam within the Schwarzschild objective. All experiments are conducted at the Laser-Laboratorium Göttingen e.V. (Göttingen, Germany).

## 3. Theory

The essential concepts and applications of the combined atomistic-continuum model MD-TTM to study the evolution of metallic solids excited with an ultrashort laser pulse are explained in Ref. [31]. The model consists of two parts: the atomic motion is described within the MD approach, whereas a diffusion differential equation for electrons is accounting for the effect of free carriers via their temperature dynamics. Thus, the combined model provides a detailed atomic-level description of the kinetics of fast non-equilibrium processes of laser melting at the atomic resolution and, at the same time, in continuum ensures for the adequate description of the laser light absorption by the conduction band electrons, the energy transfer to the lattice due to the electron–phonon coupling, and the fast electron heat conduction. Based on the atomistic-continuum approach, the MD-TTM model was further developed for simulation of the nanostructuring processes on metal targets in multiprocessing algorithm (running in parallel) to be applied in large scale modeling tasks. The model applied here and its parametrization for the investigation of UV ultrashort pulse generation of periodic pattern on metal surfaces in a high fluence regime is described in detail in Ref. [37].

The initial structures represented by supercells of (40 x 270 x 200) nm^3^ and (40 x 500 x 200) nm^3^ in X, Y, and Z directions correspondingly were prepared and consisted of 125,000,000 and 230,000,000 atoms, respectively. The MD supercells were equilibrated at room temperature before exposure to a single 1.6-ps laser pulse of wavelength equal to 248 nm with a spatially sinusoidal intensity profile with periods of 270 nm and 500 nm for the smaller and the larger sample, correspondingly. Then, the prepared computational cells were run across 256 and 480 processor cores, respectively, to simulate the nanostructuring process up to 1.5 ns of the experimental time.

The geometry and boundary conditions of the computational supercell were the same as those reported in Ref. [37], whereas, depending on the investigated feature, various techniques of the visualization and numerical analysis of the results from the point of thermo-dynamical properties were used. Thus, monitoring of the pressure and lift-off dynamics at the same time allows for extracting the mechanism of the nanostructures’ formation, whereas the simultaneous observation of the electronic and lattice temperatures explains their fast solidification [25]. Additionally, depending on the particular conditions of laser-induced phase transition processes (ultrafast melting and resolidification), local order parameter (LOP) [31] or central symmetry parameter (CSP) [40] have been used to identify the local crystal structure. The former parameter, due to its high sensitivity is frequently utilized for tracking down the ultrafast laser-induced melting process under conditions of strong superheating and high pressure and temperature gradients, whereas the latter is independent on crystal orientation and therefore allows for convenient identification of the forming nanocrystals during the solidification process. Such an approach has been in particular successfully used during the procedure of direct comparison of the simulation results with the obtained nanostructures in the experiments. Thus, the local crystal structures are readily highlighted by LOP or CSP values during the visual analysis of the atomic configurations.

## 4. Results and Discussions

Figure 2 displays scanning electron micrographs (SEM) taken from a gold surface irradiated with a 350-nm-periodic intensity profile (single pulse at 248 nm, 1.6 ps). At low fluence (a) some slight elevations at the lines of maximum fluence are visible (surface swelling). Increasing the fluence (b), these elevations become more pronounced and open up at some points forming holes (void formation). At further increased fluence (c), the elevations turn into grooves by opening up over the whole length and forming side walls. In some regions the side walls of neighboring grooves are well separated, in other regions the walls of neighboring grooves merge to form one nanoridge with some droplets on top (wall formation). Taking an even higher fluence (d), the line pattern blurs giving the impression of broad melting.

In the case of a larger structure period of 500 nm, the neighboring features at the high fluence lines are a bit more separated. Figure 3a displays the 3D simulated atomic snapshots taken 1 ns after the laser pulse absorption for three different fluences: at 130 mJ/cm^2^ slight surface elevations are visible. At 160 mJ/cm^2^ these voids are much larger, and at 250 mJ/cm^2^ the voids open leading to a rather shallow groove with steep side walls. In Figure 3b this simulation result is directly compared to the corresponding experimental data. There is a wide agreement; even details like droplets on the walls and spikes on the ground are found in both representations. Only the thickness and the height of the walls seem to deviate in the simulation from the experimental observation. While further dynamics of molten material is not excluded, a better agreement may also be reached applying alternative models for material parameters. For example, the electron-phonon coupling parameter is heavily under discussion [41,42,43]. Here, we have applied the result of Lin et al. [44], while recent calculations have yielded a lower value and showed a strong dependence on density [45]. In addition, the model for the electron heat conductivity has a strong impact on the resulting laser energy deposition profile. In the present work it was described as a function of electron and lattice temperatures [46], whereas an alternative approach accounting for the density of matter can be utilized [47]. Thus, as a result of different descriptions of the above quantities, the effective penetration depth of the laser-deposited energy [48] (was identified here as 150 nm) can deviate as well as the profile and magnitude of the established temperature and pressure gradients [16]. The relaxation of the laser-induced stresses, depending on the temperature distribution and the peak position of tensile forces, results in nucleation of the internal voids (onset of the spallation process) at a certain depth, therefore, determining the thickness of the spalled layer as a result [49].

Careful analysis of the generated structures presented in Figure 3 indicates that their formation is not due to a volume increase but mainly due to the formation of voids below the surface. Note, that there is no significant amount (or even complete absence) of free vapor atoms inside the voids, which would be necessary for generation of high internal pressure due to massive vaporization of the metal atoms and the subsequent voids growth, as it was proposed in Ref. [50]. Instead, the formation of voids, as a predecessor of the surface layer spallation, originates from the relaxation of the laser-induced stress accumulation resulting from the fast heating under the constant volume of the irradiated target. Not only this relaxation drives the already melted matter into the hydrodynamical motion upward, but also results in a mechanical rupture of the subsurface layer (spallation) in the area where the lattice temperature and pressure are high enough [16,18,32,49,51,52]. This mechanism allows for generation of arrays of subsurface voids of specific size for their technological applications [53].

Eventually, the high linear momentum of the upward motion of the solidifying structure results in a propagation of cracks along the X direction of the forming nanostructure. This is indicated by arrows in the Figure 4a at the last atomic snapshot at 1000 ps. The development of the crack leads to a complete detachment of the nanostructure tip resulting in spallation, leaving an opened hole as it is experimentally observed, see Figure 2b. The tip detachment is obvious in the analysis of the velocity lift-off field where a clear jump in its value is observed at the crack position. Interestingly, the spalled tip in Figure 4a is already in the solid phase, which is supported by the CSP analysis (CSP < 0.11) and the tip temperature has a value of 950 K, whereas the equilibrium melting point of the modeled Au material is 1337 K.

It can be clearly seen that in the non-damaged parts of the surface (more precisely: in the areas where the fluence is below the threshold) the original morphology remains intact even at the highest fluence for the case of a 500-nm period (Figure 4a). However, as the incident fluence increases (Figure 3a), or at a smaller structure period of 270 nm, this non-influenced zone becomes very narrow. This is observed in Figure 4b, where the high incident fluence of 250 mJ/cm^2^ results in an active formation of subsurface voids that develop fast and open to the surface by the time of 700 ps. While the non-influenced zone is getting narrow, the walls of neighboring grooves will collide and form one merged wall (nanoridge), as it experimentally can be seen in Figure 2c. This can also be concluded from Figure 4c, where the tangential component of the local velocities of the material in Y direction is plotted as a color field. Specifically, at the moment of the separation of the left and right wings of the growing structure, the surface tension forces add up to the tangential momentum for their further motion and the left wing has a negative Y component (−20 to −40 m/s), whereas the right wing has a positive one (+20 to +40 m/s). This means that the walls are moving outwards approaching their merging point in between the grooves. In fact, already upon the beginning of the spallation process, the side walls of the forming structure already have the tangential component of the velocity due to outward motion of the melt driven by the unloading pressure wave. At 1.5 ns after the pulse, the walls of liquid material are still moving outwards without any resistance and they will collide afterwards. Note that the smaller the period, the lower the required fluence at which merging of side walls occurs. This is supported by the experimental measurements (see Figure 5) where at a period of 270 nm, the incident fluence of 300 mJ/cm^2^ is sufficient to lead to single walls formation between the grooves, whereas it does not happen yet below the incident fluence of 400 mJ/cm^2^ at a 500-nm period. In Figure 5a–d on top of the nanoridges, droplets are forming at some irregular positions, caused by instabilities, they are also visible forming already in the MD simulation in Figure 4c after 1500 ps on the tip of the filament. The formation of droplets occurs due to Rayleigh–Plateau instability and minimizes the surface tension of the elevating and elongating wall-like structures.

Figure 6 shows the direct comparison of the modeling results with the experimental observations for the case of a 500-nm period structures formation at an incident fluence of 250 mJ/cm^2^. One can see that the final structures are opened to the surface (a) and the simulation is reproducing this process (c). Amazingly, not only the formation of the wall-like structures is observed in both simulations and experiments (green lines relate the experimental observation with corresponding cross sections in the simulated structure), but even the tiny surface and subsurface nanograins, nanovoids, nanowalls, nanocolumns, and polycrystalline structures at the base of the modeled structures reveal a perfect match between the simulation and experiments. Namely, the process of spallation creates an excessively rough surface with nanostructures and nanovoids, captured inside the solidified material (b). To make it clear, the corresponding nanocolumn (orange rectangle) and nanovoid structures (red rectangle) are identified and boxed in both the TEM (a) and FIB (b) pictures and related to the atomic snapshot (c). The formation of such structures perfectly agrees with recent theoretical findings presented in [54,55].

Apart of the generation of a great number of dislocation planes, highlighted by a light blue color in Figure 6c, the results of the simulation also predict the polycrystalline character of the solidified structures as it was emphasized in [56]. As it was found in [25], the fast cooling process of the developing structure is provided by the strong electronic heat conduction mechanism. Since the classical solidification is governed by the diffusion process, which takes time, the liquid can undergo a significant cooling below the equilibrium melting point by more than 20% [57]. Thus, at such strong undercooling conditions, the classical solidification process governed by the heterogeneous mechanism (propagation of the well-defined solid–liquid interface) cannot keep up due to a too strong cooling rate and the solidification process turns into a massive nucleation of the solid phase inside the liquid due to the homogeneous nucleation mechanism. This results in the formation of nanograins of the solid phase that have random orientation forming nanograin boundaries constituting the polycrystalline character of the final nanostructures. Moreover, the stronger the cooling rate, the smaller the resulting nanograin size. Since the cooling rate depends on the established temperature gradient, it therefore depends on both the physical size of the excited volume and the applied incident fluence, see Figure 7.

The analysis of the generated crystal structures, presented in Figure 7a–d, indicates their polycrystalline character, where mainly small nanocrystals are constituting the nanostructures generated at a smaller period and at a smaller incident fluence. The last figure in (d) allows for a direct comparison of the simulated bottom and subsurface structure generation, shown in an orange rectangle region in Figure 6a–c. The noticeably larger size of the nanocrystals is explained by a relatively weaker cooling rate limited to the nearly 1D case due to the elongated geometry of the pillar-, column- or wall-like structures. The detailed analysis of the experimental picture (Figure 6b) indicates the polycrystalline character of the subsurface region via darker lines and spots due to different optical properties of the nanograin boundaries and the relative defects. While the experimentally observed and numerically simulated nanograin structures are in agreement with recent theoretical works [49,58,59], it is worth mentioning that the crystal structures obtained during the nanostructuring process due to extremely fast laser-induced cooling mechanism can be of unique character and never met in nature [58]. Thus, generation of such structures can be of great interest for potential technological applications.

Figure 8 illustrates that by increasing the incident fluence to a higher value of 500 mJ/cm^2^ (well above the ablation threshold ~210 mJ/cm^2^), the melting extends over the whole irradiated area (40 × 40) μm^2^, and the contrast of the resulting surface pattern strongly diminishes (500-nm period) or totally vanishes (270-nm period). We refer to this observation as broad melting. In order to follow the development of an ultrafast melting process, induced in the material at such strong heating, we used here the LOP parameter that is very sensitive to the crystal structure disordering and allows for a reliable identification of the liquid phase [31,33]. Thus, in both Figure 8a,c, a well distinguished melting front due to propagation of a solid–liquid interface referred to as heterogeneous melting is not observed. Instead, due to significant overheating the phase transition proceeds by a massive nucleation of liquid phase inside the crystal referred to as homogeneous mechanism of the phase transition. Additionally, a number of dislocation planes, generated due to rapid expansion of the most heated material volume, is less resistant to the phase transition. This provides an additional input to the nucleation sites for the liquid phase, visible as red lines inside the solid phase shown in yellow/orange.

The experimental as well as the simulation results show that in addition to the laser induced upward motion of material, a lateral displacement of material plays an important role in the structure formation process. This observation is in agreement with a recent experimental and theoretical work on redistribution of the processed material volume during the femtosecond laser processing [51]. In contrast to material irradiation by a single focused beam to a small spot where the material can expand freely in lateral directions, in the case of a periodic lateral pattern, the material expulsions from neighboring grooves at high fluence affect each other. This leads to the observed specific features like steep ridges at a moderate incident fluence and a flatter pattern at higher energy deposition values.

Finally, the good agreement between the modeling results and the experimental data justifies the proposed approach as a powerful tool revealing the physics behind the nanostructuring process at a gold surface and providing a microscopic insight into the dynamics of the structuring processes of metals in general. Moreover, the results discussed above are applicable to other metals as well, whenever fast and localized laser energy deposition occurs. In particular, transition metals such as Al and Ni, due to a shorter characteristic distance (75–125nm as compared to Au 150–200nm) of the laser-deposited heat penetration [48] (heat affected zone), could be better candidates for the achievement of cleaner structures at a shorter structure period. Reducing the structure period below this distance, however, can deteriorate the resulting surface quality due to laser-deposited dissipated heat (governed by the fast electron heat conduction process) and therefore smoothing the resulting pattern.

## 5. Conclusions

The performed experiments of periodic patterning with a single UV ultrashort laser pulse and corresponding simulations with the combined atomistic-continuum model allowed us to draw the following conclusions about the mechanism responsible for the nanostructures formation: the outward hydrodynamic motion of the expelled melted surface layer and its subsequent opening result in a tangential component of the velocity and lead to merging of neighboring walls of the forming structures and the formation of nanoridges on the target surface. The complete solidification of the forming structures at the fluencies well above the ablation threshold as well as the solidification of the generated droplets takes much longer (tens of ns) time scale. The strong cooling rate of the forming structures is responsible for the formation of the nanofeatures with a polycrystalline character constituting the nanosize features of various shapes. The broad melting observed at high fluencies was identified as a reason for diminished contrast of the generated structures as a result of the molten material redistribution. The applied numerical model has been identified as a powerful numerical tool for the investigation of the ultrashort laser pulse nanostructuring process on metals. The results of the performed study can have a strong impact on potential industrial applications for the generation of material surfaces with predesigned properties and topography.

## Figures and Tables

**Figure 1 nanomaterials-10-01998-f001:**
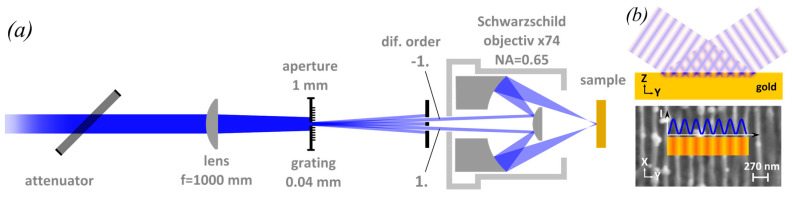
(**a**) Mask projection setup for periodic surface structuring. A sinusoidal interference pattern is obtained by using only the ± 1st diffraction orders for mask imaging. (**b**) Shape of the intensity distribution generated by two beam interference.

**Figure 2 nanomaterials-10-01998-f002:**
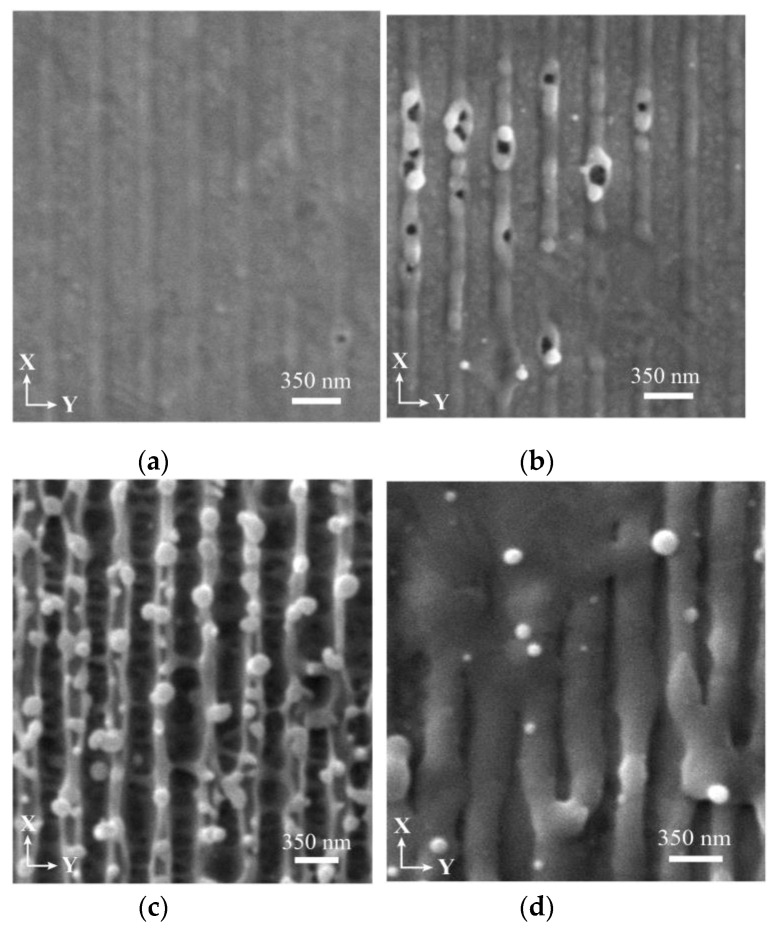
Surface topography obtained on gold after single pulse irradiation with a 350-nm period. (**a**) Surface swelling at 100 mJ/cm^2^, (**b**) void formation at 150 mJ/cm^2^, (**c**) wall formation at 200 mJ/cm^2^, (**d**) broad melting at 350 mJ/cm^2^.

**Figure 3 nanomaterials-10-01998-f003:**
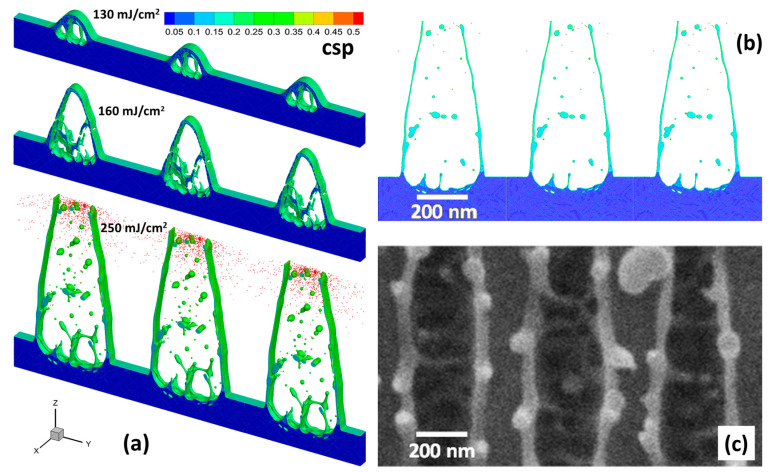
(**a**) Simulated 3D atomic snapshots 1 ns after 1.6 ps single pulse irradiation with a structure period of 500 nm. Fluence: 130 mJ/cm^2^ (top), 160 mJ/cm^2^ (middle), 250 mJ/cm^2^ (bottom). The atoms are colored by central symmetry parameter (CSP) for distinguishing a local crystal structure as follows: crystal < 0.08 < defects (dislocations) < 0.11 < liquid < 0.25 < surfaces < 0.50 < vapor (free atoms). (**b**) Direct comparison of the simulation (cross section for the case of a 500-nm period) with the (**c**) experimental result (SEM, top view) on the same length scale at the incident fluence of 250 mJ/cm^2^.

**Figure 4 nanomaterials-10-01998-f004:**
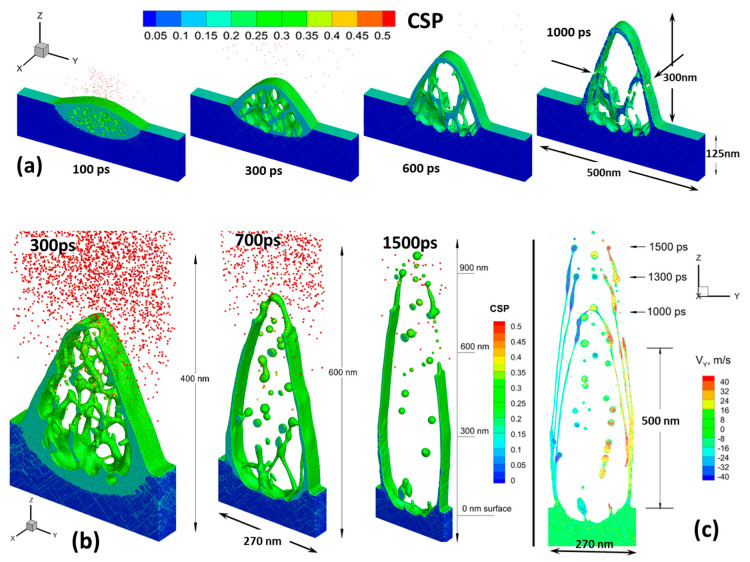
(**a**) Sequence of atomistic snapshot configurations (100 ps, 300 ps, 600 ps, and 1000 ps) of a 500-nm period structures obtained at the incident fluence of 160 mJ/cm^2^. The arrows on the 1000 ps snapshot indicates the formation of a crack along the nanostructure direction X. (**b**) Calculated shapes for 300 ps, 700 ps, 1.5 ns after the pulse for a 270-nm period structure obtained at an incident fluence of 250 mJ/cm^2^. The atoms are colored by CSP for distinguishing a local crystal structure as follows: crystal < 0.08 < defects (dislocations) < 0.11 < liquid < 0.25 < surfaces < 0.50 < vapor (free atoms). (**c**) The tangential (perpendicular to the incident pulse) Y component of the velocity field of the growing structure in (**b**) is shown by color for the set of times: 1000 ps, 1300 ps, and 1500 ps.

**Figure 5 nanomaterials-10-01998-f005:**
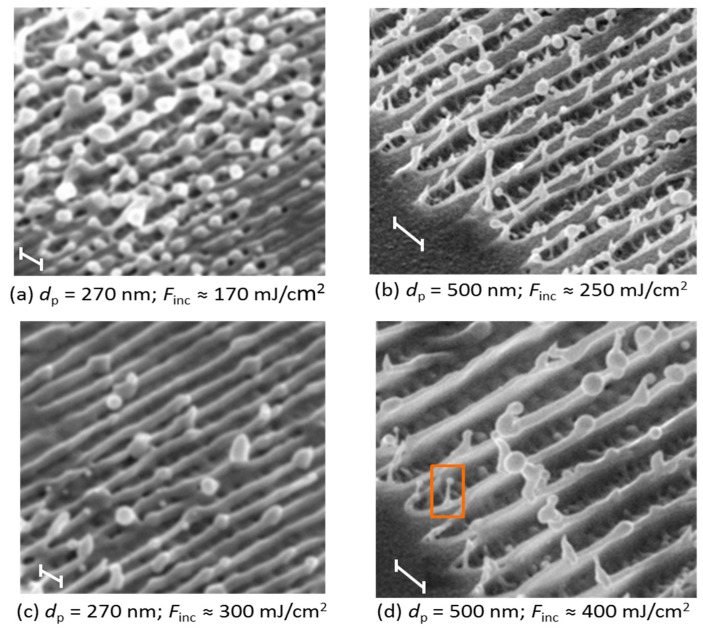
Periodic patterns obtained with periods of 270 nm in (**a**) and (**c**) and 500 nm in (**b**) and (**d**) at various fluences, at a tilt angle 45°, the period is indicated by the scale bars. For larger periods, a higher fluence is required to obtain nanoridges composed of two collided walls. The squared structure in (**d**) is experimentally identified and obtained in simulations (Figure 6).

**Figure 6 nanomaterials-10-01998-f006:**
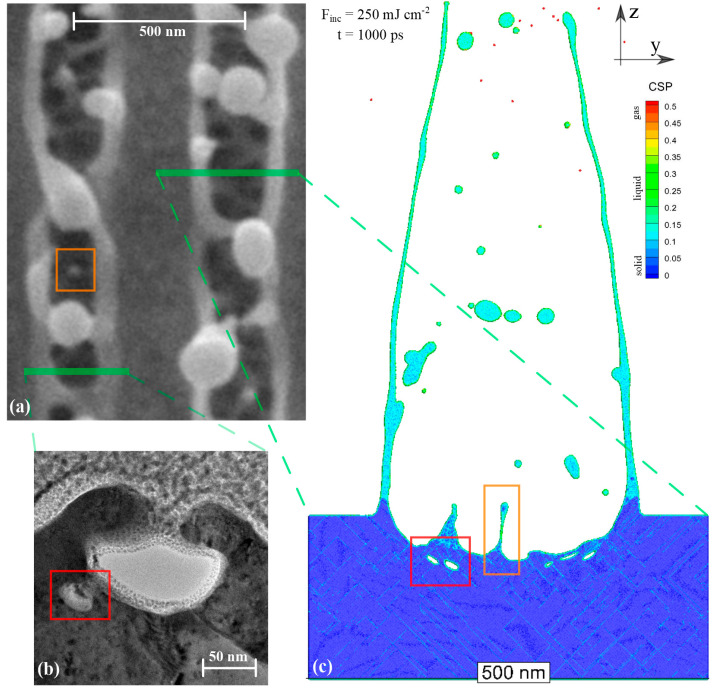
Direct comparison of SEM top view in (**a**) and TEM cross-section in (**b**) with the simulation results of the atomic configuration 40 x 500 x 200 nm^3^ of a 500-nm period at 1000 ps in (**c**). The atoms are colored by CSP parameter. The process of spallation results in an opening of the generated structure and formation of a number of droplets and pillar-like structures on the remaining surface (orange rectangle). The spallation process is accompanied with the generation of subsurface voids (red rectangle) registered in both experiment and simulation and is magnified in the squared window in Figure 7d below.

**Figure 7 nanomaterials-10-01998-f007:**
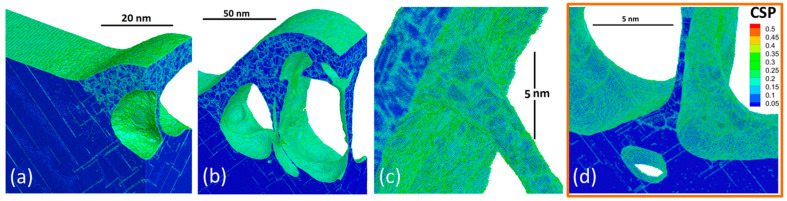
The nanostructures generated in the simulations on periodic nanostructuring are magnified for (**a**) the period of 270 nm at the incident fluence of 120 mJ/cm^2^ and (**b**) 145 mJ/cm^2^. (**c**) Structures obtained for the case of a 500-nm period at the incident fluence of 160 mJ/cm^2^ and (**d**) 250 mJ/cm^2^. The atoms are colored by CSP parameter.

**Figure 8 nanomaterials-10-01998-f008:**
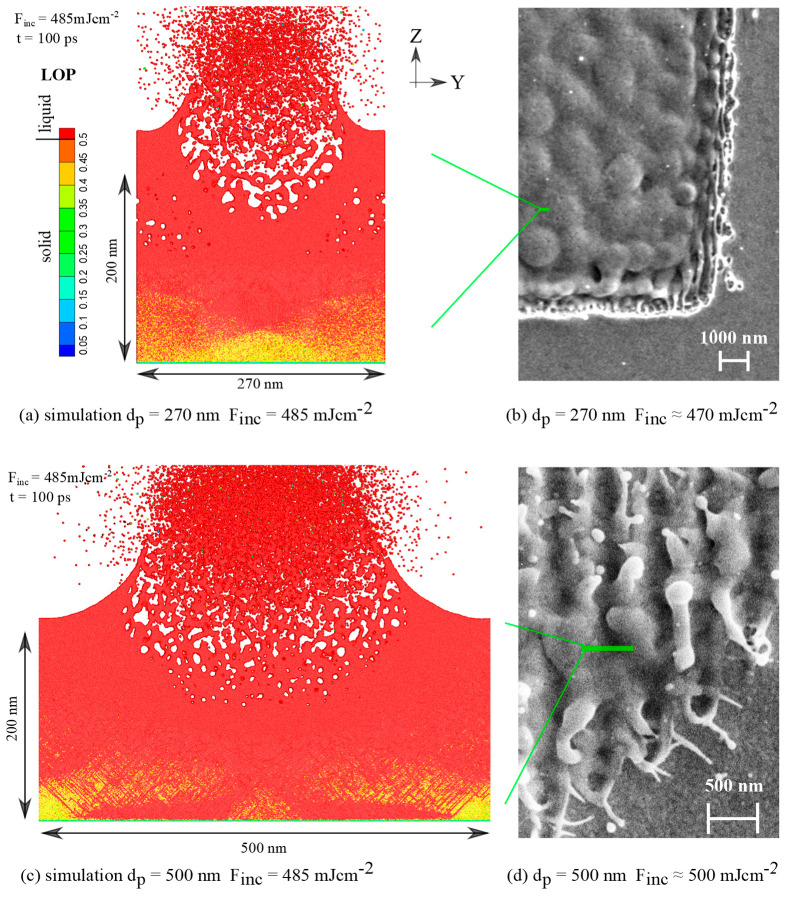
(**a**) Direct comparison of the simulation results of 270-nm period nanostructuring at a high incident fluence of 485 mJ/cm^2^ with (**b**) the experimental data obtained for 470 mJ/cm^2^. (**c**) Direct comparison of the simulation results of 500-nm period nanostructuring at a high incident fluence of 485 mJ/cm^2^ with (**d**) the experimental data obtained for 500 mJ/cm^2^. The atoms are colored by local order parameter (LOP) used here for identification of the extremely fast melting process. The liquid phase is characterized by a rapid jump of LOP above cut-off value 0.5.

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
