# Peer review of "Formation of Periodic Nanoridge Patterns by Ultrashort Single Pulse UV Laser Irradiation of Gold"

_nanomaterials, 2020, doi:10.3390/nano10101998_

Round 1
Reviewer 1 Report
Authors in their manuscript nanomaterials-939127 titled "Formation of periodic nanoridge patterns by ultrashort UV laser irradiation of gold" present novel results of the experiment and atomistic modeling on periodical interference nanostructuring by single-pulse irradiation. The results are very novel, well-conducted, and beautify presented, thus definitely deserves publication. However, several problems are spotted in the current form. Thus, major revision is recommended before publication in the Nanomaterials journal by MDPI.
- The authors use atomistic modeling to calculate gold film behavior after interference picosecond pulse irradiation. The thermal diffusion influence for structure quality in laser interference ablation has recently been studied in [DOI: 10.1039/C7CP08458G] for silicon. I recommend providing an estimation of how results presented in this paper (270 nm, 350 nm, and 500 nm periods and 1.6 ps pulse duration) correspond to thermal modulation depths and qualities of structures for the gold film.
- This interference experimental setup provides a periodical intensity distribution profile on irradiated materials. Such kind of setup using nanosecond laser has already been used for periodic pattern formation also of nanoparticle redistribution [DOI: 10.2961/jlmn.2012.03.0022]. I would recommend including a nanoparticle size distribution chart into the paper.
- The micro and nanobubbles on the in the molten and re-solidified cylindrical ridge in the laser processed might have occurred because of Rayleigh-Plateau instability and Marangoni effect [DOI: 10.1016/j.apsusc.2013.03.092]. This instability is related to surface tension, and particular small random oscillation growth to measurable periodical oscillation. The discussion related to the origin of the nanosphere formation in the ridge formation should be reconsidered. I recommend include some possible mechanisms to the dissection. The Marangoni effect is mentioned in the introduction. However, Rayleigh-Plateau instability is not mentioned. I would recommend the authors consider introducing Rayleigh-Plateau instability as one of the physical mechanisms of beak up of the ridge into periodical or quasi periodical nano-bubbles.
- The experimental results and atomistic modeling of periodical structuring using single pulse exposure have already been reported [DOI: 10.1007/s12274-020-2852-3; DOI: 10.1039/D0NR00269K]. I recommend including some wider literature review dedicated to the periodical nano-structuring using single pulse irradiation.
- The caps letter has to be replaced by the small letter "formation: The upward" as "formation: the upward" in the CONCLUSIONS section.
- Please indicate at what tilt angle SEM micrographs are taken in Figure 5.
- The text size in Figure 1 (b) "gold" and "270 um" has to be increased in order to be more readable.
- The double-space" " sign has to be removed form reference [38].
- The "Appl Sur Sci" has to be replaced by "Appl Surf Sci" in references [5] and [33].
- The journal name "JLMN-Journal of Laser Micro/Nanoengineering" has to be replaced by the abbreviation "J. Laser Micro Nanoen." in references [10] and [19].
- I would strongly recommend incorporating a "single pulse" into the title of the manuscript.
- The word "nanoridge" used in the title of the manuscript, however, is not found anywhere in the text of manuscript. I would recommend removing the nano-ridge from the title, or use it more frequently in the text of the manuscript.
13. Authors claim that they work in below ablation threshold "the areas where the fluence is below the threshold" (page 7). However, the authors do not provide an ablation threshold for the gold film. The ablation threshold evaluation of Au film and UV irradiation using 1.6 ps pulse is needed because it is not found in the text of the manuscript. There is a special technique called the D2 versus Fluence method [DOI: 10.1038/s41598-018-35604-z] for direct measurement of laser ablation threshold. I recommend evaluated the ablation threshold of the Au target or at least give literature values for a similar picosecond UV type of irradiation.
Reviewer 2 Report
The work is clear and the results are very interesting, therefore my opinion is that it is ready for its publication
Author Response
The authors thank the reviewer for this positive evaluation of the manuscript.
Reviewer 3 Report
The article by Blumenstein et al. reports on the computer simulation of structure formation on gold surfaces upon laser irradiation with UV femtosecond pulses through different gratings. The simulation was based on a hybrid atomistic-continuum model previously exploited before which separate the contribution of the atoms from that of the electrons. The comparison between simulation and experiments is quite impressive and the approach allows obtaining precious and precise information about the underlying physics of the structure formation on gold by femtosecond laser pulses. The paper is clearly written and the results can be of great interest for a relatively broad community working in surface nanostructuring.
Author Response

(The authors gave the same response as above.)

Reviewer 4 Report
In this manuscript, the authors present an experimental and theoretical study on the formation of periodic nano ridge patterns by ultrashort UV laser irradiation of gold. The micro- and nano-patterns produced upon single pulse irradiation is analyzed by scanning and transmission electron microscopy (SEM and TEM). Complementarily, hybrid atomistic-continuum model simulations have been carried out for understanding the mechanisms operative in the nano-structuring process. The formation of narrow ridges composed of colliding sidewalls due to the laser pulse interaction with the target (gold in this case) could be described.
The writing style is right, and the introduction section and method details are correct. Previously published results are introduced. I would suggest that the previous studies should be more critically presented, indicating expressly what has been achieved and what is missing. The compelled experimental and theoretical approach described in the paper makes the manuscript much more robust and understandable.
As minor requests, I would suggest the authors comment on the following points:
1.- Could other targets be tested to identify better candidates to avoid the defects and irregularities observed using gold?
2.- Which particular problems should this technique face when the patterns scale in the low 10-50 nm range?
In summary, the manuscript does help to understand the essential issues of the current production of periodic nano ridge patterns. In this general context, the paper can be accepted in the present form, with minor corrections.
Round 2
Reviewer 1 Report
Authors in their revised manuscript nanomaterials-939127 titled “Formation of periodic nanoridge patterns by ultrashort single pulse UV 2 laser irradiation of gold“ present novel state-of-the-art results of periodic structuring using ultrashort single pulse irradiation of gold. The authors have performed a good job by revising the manuscript in a short period of time according to most of the comments indicated by the reviewers. The quality of the manuscript has improved by a big margin; thus, I strongly recommend publishing it by the Nanomaterials journal by MDPI.